# Sports safety matting diminishes cardiopulmonary resuscitation quality and increases rescuer perceived exertion

**Thomas Kingston[1]ᵒ, Nicholas B. Tiller[2], Elle Partington[3], Mukhtar Ahmed[4], Gareth Jones[5], Mark I. Johnson👤[5], Nigel A. Callender👤[5,6]ᵒ ***

1 Department of Anaesthesia and Critical Care, Barnsley District General Hospital, Barnsley, South Yorkshire, United Kingdom, 2 Institute of Respiratory Medicine and Exercise Physiology, Lundquist Institute for Biomedical Innovation at Harbor-UCLA Medical Centre, Torrance, California, United States of America, 3 Newcastle Medical School, Newcastle University, Newcastle Upon Tyne, Tyne and Wear, United Kingdom, 4 Emergency Medicine Department, Leeds Teaching Hospitals, Leeds, West Yorkshire, United Kingdom, 5 School of Clinical and Applied Sciences, Leeds Beckett University, Leeds, West Yorkshire, United Kingdom, 6 Department of Anaesthesia, Northumbria Specialist Emergency Care Hospital, Cramlington, Northumberland, United Kingdom

ᵒ These authors contributed equally to this work.
* n.callender@leedsbeckett.ac.uk

**Data Availability Statement:** All relevant data are within the manuscript and its Supporting information files.

## Abstract

### Objectives

Compliant surfaces beneath a casualty diminish the quality of cardiopulmonary resuscitation (CPR) in clinical environments. To examine this issue in a sporting environment, we assessed chest compression quality and rescuer exertion upon compliant sports safety matting.

### Methods

Twenty-seven advanced life support providers volunteered (13 male/14 female; mass = 79.0 ± 12.5 kg; stature = 1.77 ± 0.09 m). Participants performed 5 × 2 min, randomized bouts of continuous chest compressions on a mannequin, upon five surfaces: solid floor; low-compliance matting; low-compliance matting with a backboard; high-compliance matting; high-compliance matting with a backboard. Measures included chest compression depth and rate, percentage of adequate compressions, and rescuer heart rate and perceived exertion.

### Results

Chest compression depth and rate were significantly lower upon high-compliance matting relative to other surfaces ($p<0.05$). The percentage of adequate compressions (depth ≥50 mm) was lowest upon high-compliance matting (40 ± 39%) versus low-compliance matting (60 ± 36%) and low-compliance matting with a backboard (59 ± 39%). Perceived exertion was significantly greater upon high-compliance matting versus floor, low-compliance matting, and low-compliance matting with a backboard ($p<0.05$).

**Funding:** No financial support was received in relation to this study. Temporary loan of equipment was received from Laerdal Medical UK.

**Competing interests:** This study received the temporary loan of equipment from Laerdal Medical UK. NAC is the owner of a commercial indoor climbing gymnasium. NBT is funded by a postdoctoral fellowship from the Tobacco-Related Disease Research Program (TRDRP; award no. T31FT1692). MIJ reports grants from Glaxosmithkline and the Neuromodulation Society of the United Kingdom and Ireland (NSUKI) including consultancy fees from TENSCare, outside the submitted work. This does not alter our adherence to PLOS ONE policies on sharing data and materials. There are no other conflicts of interest to declare.

## Conclusion

Providers of CPR should be alerted to the detrimental effects of compliant safety matting in a sporting environment and prepare to alter the targeted compression depth and rescuer rotation intervals accordingly.

## Introduction

Global survival rates following cardiac arrest are low (2.8–47.4%) [1] and influenced by factors including the quality of cardiopulmonary resuscitation (CPR) and automated defibrillator (AED) access [2]. Although cardiac arrest during exercise is rare, with an estimated incidence of 4.5 per million per year [3, 4], sport and recreation venues have been identified as higher-risk environments [5]. In such venues, however, prompt bystander-delivered CPR and AED deployment are more likely and, therefore, confer a better outcome [6, 7].

Chest compression depth and rate, two primary indices of CPR quality, partly determine outcome in cardiac arrest. Importantly, chest compressions should be performed upon a firm surface, and achieve a depth of 50–60 mm at a rate of 100–120 compressions per minute [2]. Achieving these targets may be influenced by many factors in a sports setting, including the surface present beneath a casualty, a factor which has been found to attenuate chest compression quality in clinical settings [8–10].

Sporting facilities are a non-clinical environment where compliant undersurfaces are frequently encountered (e.g. gymnastics facilities and martial arts gymnasiums). Indeed, some venues such as indoor climbing gymnasiums may feature very large expanses of floor area confluently overlaid with high-compliance matting. In these environments, considerable and/or unacceptable delay may be incurred while relocating a casualty to a firm surface upon which to perform chest compressions. Despite safety matting being commonplace in the sports and leisure industry, the influence of such surfaces on CPR quality has, to our knowledge, not yet been investigated.

To facilitate the development of future guidance on CPR delivery in sporting facilities which feature compliant matting surfaces, the aims of this study were: i) to ascertain the effect of compliant sports safety matting upon markers of chest compression quality and rescuer exertion, and ii) to investigate whether the use of a CPR backboard attenuated any detrimental effects of a more compliant surface.

## Methods

### Participants

Thirty-two advanced life support (ALS) providers (13 male/19 female; age: 30.1 ± 6.8 y; mass: 75.7 ± 14.1 kg; height: 1.75 ± 0.10 m; mean ± SD) were recruited from local paramedic (n = 10 participants), critical care (n = 6) and emergency medicine (n = 16) departments. Using previous work by Perkins et al. [8] and Tweed et al. [9] a power calculation was performed *a-priori* (G*Power, Dusseldorf, Germany; α: 0.05, β: 0.8, two-tailed), anticipating a medium effect size for the primary dependent variables (partial eta squared; $\eta^2_p = 0.06$). Employing a Repeated-Measures Analysis of Variance (ANOVA), n = 26 was estimated as sufficient. Leeds Beckett University ethical review board granted approval, and participants provided written, informed consent. All participants were free from known pre-existing cardiorespiratory illness (medical

questionnaire) and medications known to affect the cardiovascular response to exercise. Before testing, participants were asked to abstain from alcohol and caffeine for 24h.

## Experimental protocol

The effects of matting compliance on compression quality were assessed using a randomized, single-blind, crossover design at a commercial indoor climbing gymnasium. Participants delivered chest compressions on a training mannequin beneath which were matting surfaces of varying compliance. Measures of chest compression quality and rescuer exertion were recorded throughout.

## Procedures

Following 10-min of quiet sitting, baseline telemetered heart rate (HR; Polar T31, Polar Electro, Finland) and rating of perceived exertion (RPE; CR-100 scale; [11]) were recorded. Prior to testing, a single 20 s bout of compressions was performed on the resuscitation mannequin with visual feedback from the software in an attempt to ensure consistent performance among participants. Following a brief rest, participants then performed $5 \times 2$-min bouts of continuous, compression-only CPR upon various surfaces (Table 1) in a randomized order, each separated by a 5-min seated rest period (Fig 1). Participants were instructed to adhere to the 2015 ERC resuscitation guidelines (Compression rate: 100–120 min$^{-1}$; depth: 50–60 mm; [2]).

## Measures

Chest compressions were performed on a commercially available resuscitation training mannequin (Resusci-Anne QCPR; Laerdal Medical, Norway), incorporating an electronically-metered internal depth gauge and recoil spring. To achieve a torso mass equivalent to a 75 kg male phantom (32.6 kg; [12]), lead sachets were secured within the thorax compartment of the mannequin.

**Compression data.** Data on each compression were recorded automatically by the mannequin's proprietary software (SkillReporter PC, V 3.2.0.1; Laerdal Medical, Norway) for exactly 2-min, initiated at the first compression. Maximum and minimum compression depth (mm) was recorded for each compression (compression depth and depth at end of release). Correspondingly, the percentage of compressions attaining the target depth threshold ($\geq$50 mm) were recorded. Mean compression rate (min$^{-1}$) was calculated from the inter-

**Table 1. Descriptions of surfaces beneath the mannequin during each compression bout.**

| Surface | Abbreviation | Description |
|---|---|---|
| Floor | *Floor* | Solid concrete floor. |
| Low-compliance foam | *LC* | The foam matting *in-situ* in the gymnasium (Vitafoam VE38 200, density: 38–40 kg/m³; hardness: 180–220 N). |
| Low-compliance foam & backboard | *LCBB* | As for *LC* but with a rigid backboard (800 mm x 500 mm x 18 mm rigid board) inserted between the manneqin and matting. |
| High-compliance foam | *HC* | Foam of higher-compliance used in some sports settings (Vitafoam VE40 400, density 40 kg/m³; hardness: 110–130 N). |
| High-compliance foam & backboard | *HCBB* | As for *HC* but with the rigid backboard. |

All surfaces except for the Floor comprised three 100 mm layers of foam laid to a total depth of 300 mm, as is standard in venues similar to that where testing was undertaken. All surfaces were covered with an opaque sheet to remove visual cues for participants. N = Newtons.

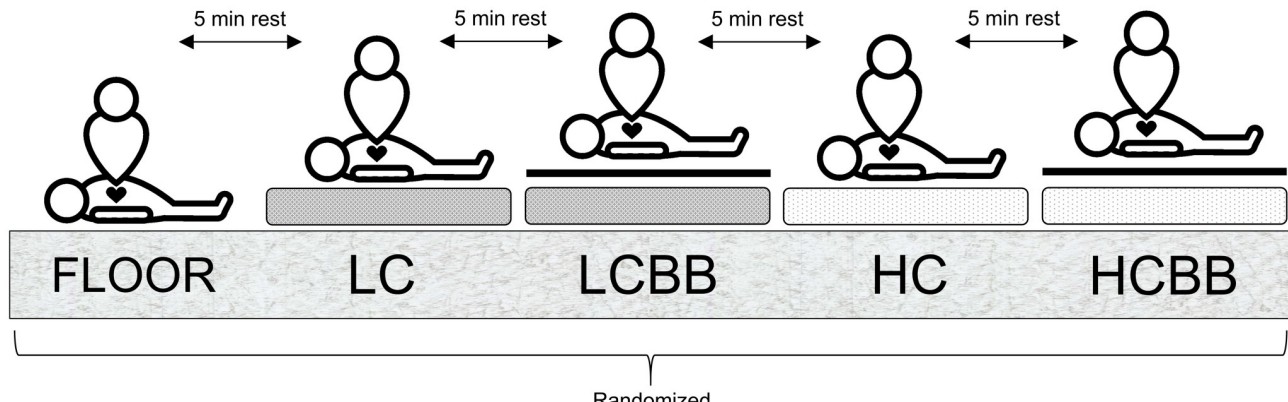

**Fig 1. Schematic of experimental protocol.** Bouts of continuous, compression-only CPR (2 min) were performed upon five surfaces in a random order: Floor = Concrete floor; LC = Low-compliance foam; LCBB = Low-compliance foam & backboard; HC = High-compliance foam; HCBB = High-compliance foam & backboard.

compression time interval. Mean compression data were reported as that averaged over the entire 2-min bout, unless specified.

**Rescuer exertion.** Participant HR was recorded during the 60 seconds prior to each trial ($HR_{Pre-Task}$) and in the final 20 seconds of compressions ($HR_{Peak}$). Perceived exertion was rated immediately before each compression bout ($RPE_{Pre-Task}$) and immediately following the 2-min epoch ($RPE_{Peak}$).

## Statistical analyses

Raw participant data were exported from the mannequin software to Microsoft Excel (Microsoft Corporation, Washington, USA) and analyzed using SPSS Version 26 (IBM, Chicago, USA). Normality of distribution was assessed using the Shapiro-Wilk test. Compression variables (compression rate, depth and percentage of adequate compressions) were compared among surfaces using a One-Way Repeated-Measures Analysis of Variance (ANOVA). Participant heart rate ($HR_{Pre-Task}$ *vs.* $HR_{Peak}$) and RPE ($RPE_{Pre-Task}$ *vs.* $RPE_{Peak}$) were analyzed using a Two-Way Repeated-Measures ANOVA. The Greenhouse-Geisser correction was applied where the assumption of sphericity was violated and all *post-hoc* analyses were performed with Bonferroni adjustment. Effect size was estimated using the partial eta-squared ($\eta^2_p$) method and categorized as small = 0.01, medium = 0.06, or large = 0.14 [13]. Data are presented as mean ± SD with critical alpha-level applied as 0.05, and confidence interval of 95%.

## Results

### Participants

From the 32 participants recruited, five subjects (all female) were excluded from the analysis owing to a failure to deliver chest compressions of the pre-determined cutoff depth (>40 mm; 80% of minimum target depth) on solid floor (mean compression depth: 34.4 ± 3.4 mm). When compared to the remaining female participants (n = 14), those excluded were of a similar age (28.2 ± 3.2 *vs.* 29.1 ± 4.0 y; $p$>0.05; $\eta^2_p$: 0.01), but exhibited a significantly lower body mass (58.0 ± 8.5 *vs.* 74.4 ± 12.5 kg; $p$ = 0.015, $\eta^2_p$: 0.30), stature (1.65 ± 0.04 *vs.* 1.71 ± 0.09 m; $p$ = 0.036, $\eta^2_p$: 0.24), and BMI (21.5 ± 3.5 *vs.* 25.3 ± 3.3; $p$ = 0.042, $\eta^2_p$: 0.22). Accordingly, 27 participants were included in the final analysis (age: 31.5 ± 7.3 y; mass: 79.0 ± 12.5 kg; 1.77 ± 0.09 m; 13 male/14 female).

## Compression data

**Compression depth.** Compression metrics among all surfaces are shown in Table 2 and Fig 2. There was a significant main effect of surface (F[2.83, 73.63] = 5.42, $p$ = 0.002; $\eta^2_p$: 0.17). *Post-hoc* analyses showed the compression depth achieved upon the *HC* surface was significantly lower than that achieved upon the *Floor* ($p$ = 0.003), *LC* ($p$ = 0.001), and *LCBB* ($p$ = 0.039) surfaces. There was no difference in mean compression depth upon the *Floor* and *LC*, *LCBB* or *HCBB* (all $p$ = 1.000); *LC* and *LCBB* or *HCBB* (both $p$ = 1.000); *HCBB* and *LCBB* ($p$ = 0.949); or *HC* and *HCBB* ($p$ = 0.786). For percentage of compressions attaining an adequate depth ($\geq$50 mm), there was a significant main effect of surface (F[2.98, 77.59] = 3.53, $p$ = 0.019; $\eta^2_p$: 0.12). *Post-hoc* analyses found that the percentage of adequate compressions was lower for the *HC* compared with the *LC and LCBB* surfaces ($p<$0.001 and = 0.033, respectively). No difference in the percentage of adequate compressions was observed between the *Floor* and *LC* ($p$ = 0.956), *HC* ($p$ = 0.767), *HCBB* or *LCBB* (both $p$ = 1.000); *LC* and *LCBB* or *HCBB* (both $p$ = 1.000); *LCBB* and *HCBB* ($p$ = 1.000); or *HC* and *HCBB* ($p$ = 1.000).

**Compression rate.** There was a significant main effect of surface (F[4, 104] = 11.47, $p<$0.001; $\eta^2_p$: 0.30; Table 2 and Fig 2). *Post-hoc* testing revealed that the *HC* and *HCBB* surfaces resulted in significantly lower compression rates relative to the *Floor* ($p<$0.001 and = 0.005, respectively), *LC* ($p$ = 0.002 and = 0.007, respectively) and *LCBB* surfaces ($p$ = 0.006 and = 0.005, respectively). No difference was observed between the *HC* and *HCBB* surfaces ($p$ = 1.000), or between the *Floor*, *LC and LCBB* surfaces (all $p$ = 1.000).

## Rescuer exertion

**Rating of perceived exertion.** Data for RPE following *post-hoc* analyses are shown in Table 3. There was a significant main effect of time (RPE$_{Pre-Task}$ vs. RPE$_{Peak}$; F[1, 24] = 89.03, $p<$0.001; $\eta^2_p$: 0.79) and surface (F[4, 96] = 11.47, $p<$0.001; $\eta^2_p$: 0.32), and a significant interaction between the two (F[4, 96] = 13.62, $p<$0.001; $\eta^2_p$: 0.36). *Post-hoc* analyses revealed that the *HC* and *HCBB* surfaces elicited a greater RPE$_{Peak}$ than the *Floor* ($p$ = 0.002 and = 0.002, respectively) and *LC* surfaces ($p<$0.001 and = 0.010, respectively), with the *HC* surface also significantly different to the *LCBB* surface ($p$ = 0.001). There was no difference between the *HC* and *HCBB* ($p$ = 1.000) surfaces or between the *HCBB* and *LCBB* surfaces ($p$ = 0.101).

**Heart rate.** Data for HR following *post-hoc* analyses are shown in Table 3. There was a significant main effect for time HR$_{Pre-Task}$ vs. HR$_{Peak}$; (F[1, 24] = 297.75, $p<$0.001; $\eta^2_p$: 0.93) but no significant main effect for surface (F[2.72, 65.31] = 1.90, $p$ = 0.143; $\eta^2_p$: 0.07). A significant surface $\times$ time interaction was also present (F[4, 96] = 87.88, $p$ = 0.018; $\eta^2_p$: 0.12). Data pertaining to absolute compression decay and chest leaning are presented in S1 File.

## Discussion

This study examined the effect of compliant safety matting on chest compression quality in a sporting environment. Our findings show that high-compliance safety matting found in sporting environments significantly reduced chest-compression quality (compression depth and rate), and increased rescuer perceived exertion. Both were partially improved with use of a backboard.

High-compliance safety matting, frequently encountered in sporting venues, significantly reduced mean compression depth by ~3 mm, and attenuated the percentage of adequate compressions by up to 20% when compared to the *Floor*, *LC* and *LCBB* surfaces. The effect of the high-compliance surface was similar to that attributed to air- and foam-filled mattresses in a clinical environment [8, 9, 14]. Although the absolute decrement in chest compression depth with the *HC* surface may be relatively small, this difference in depth is comparable to that

**Table 2. Compression metrics among surfaces.**

| | (a)Floor | | | (b)LC | | | (c)LCBB | | | (d)HC | | | (e)HCBB | | |
|---|---|---|---|---|---|---|---|---|---|---|---|---|---|---|---|
| **Depth (mm)** | 49.8 | ± | 4.3[d] | 50.2 | ± | 5.0[d] | 50.0 | ± | 5.2[d] | 47.0 | ± | 4.9[a,b,c] | 48.6 | ± | 4.9 |
| *CI (95%)* | 48.1 | - | 51.5 | 48.2 | - | 52.1 | 47.9 | - | 52.0 | 45.1 | - | 49.0 | 46.6 | - | 50.5 |
| **Rate (min$^{-1}$)** | 127 | ± | 13[d,e] | 125 | ± | 13[d,e] | 125 | ± | 16[d,e] | 119 | ± | 11[a,b,c] | 120 | ± | 14[a,b,c] |
| *CI (95%)* | 121 | - | 132 | 120 | - | 131 | 119 | - | 132 | 115 | - | 123 | 115 | - | 125 |
| **Adequate (%)** | 51 | ± | 38 | 60 | ± | 36[d] | 59 | ± | 39[d] | 40 | ± | 39[b,c] | 49 | ± | 42 |
| *CI (95%)* | 36 | - | 66 | 46 | - | 74 | 44 | - | 74 | 25 | - | 56 | 32 | - | 65 |

Data are mean ± *SD* (n = 27). Floor = concrete floor; LC = low-compliance foam; LCBB = low-compliance foam with backboard; HC = high-compliance foam; HCBB = high compliance foam with backboard. Depth = mean compression depth during 2-min bout; Rate = mean compression rate (.min$^{-1}$); Adequate = percentage of compressions achieving ≥50 mm depth during 2-min bout. CI (95%) = 95% confidence interval.

[a]Significantly different versus Floor;

[b]Significantly different versus LC;

[c]Significantly different versus LCBB;

[d]Significantly different versus HC;

[e]Significantly different versus HCBB.

Alpha level = <0.05.

differentiating survival from non-survival during cardiac arrest [15]. It is also similar in magnitude to that conferring improved survival rates to hospital admission [16] and hospital discharge [17], suggesting this decrement might still be clinically meaningful.

Compression rate was also reduced upon both the *HC* and *HCBB* surfaces relative to the *Floor*, *LC* and *LCBB* surfaces. This is again consistent with hospital studies which show reduced compression rates on higher-compliance surfaces [18]. Compression depth and rate often show an inverse association [19–21], where the reduction in rate is due to the greater vertical hand-travel required to attain deeper compression depths [10, 22], as may be the case on more compliant sports matting.

Notably, no differences were apparent between the *Floor*, *LC* and *LCBB* surfaces for compression depth or rate, suggesting effective CPR may be delivered at least upon certain lower-compliance matting types found in some sports venues. As the transport of a collapsed casualty to a firmer surface is likely to introduce an unacceptable delay to treatment [23], and in the absence of specific knowledge of the effects of all safety matting varieties, it seems prudent that rescuers performing CPR should anticipate a negative effect from sports matting, and aim to compensate appropriately.

Introducing a backboard partially attenuated the loss of compression quality on the *HC* surface, resulting in a 9% increase in the percentage of compressions achieving an adequate depth, and an increase in absolute mean compression depth of ~1.6 mm. Although these measures did not attain statistical significance, research within hospital environments has shown reductions in mattress displacement, and associated rescuer workload, when backboards were used upon compliant hospital beds [10, 22]. No difference in compression depth or the percentage of adequate compressions occurred when a backboard was used upon the low-compliance surface, and similarly no effects were observed on any surface for compression rate. Notwithstanding the above, the use of a backboard upon complaint sports surfaces needs further consideration, and where high-compliance undersurfaces are present, we would advocate backboard use if placement does not delay CPR.

Rescuer exertion is an important factor in the delivery of high-quality CPR. Relative to pre-task values, heart rate and perceived exertion increased during all 2-min bouts of chest

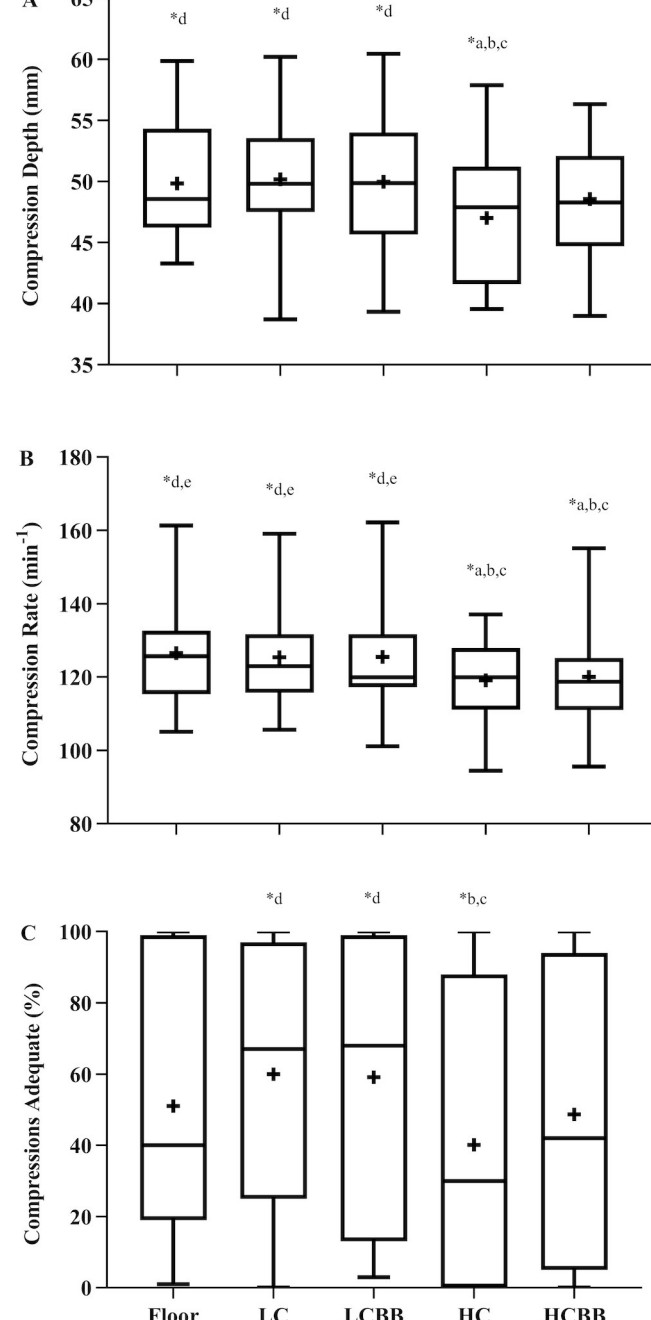

**Fig 2. Influences of matting composition on chest compression parameters.** Box and whisker plots depicting compression depth (panel A), compression rate (panel B), and percentage of adequate compressions (≥50 mm; panel C). Boxes illustrate upper and lower quartiles; whiskers = maximum and minimum values; horizontal line = median; + represents group mean value. [a]Significantly different versus Floor; [b]Significantly different versus LC; [c]Significantly different versus LCBB; [d]Significantly different versus HC.

compressions (Table 3), with the *HC* and *HCBB* surfaces eliciting values for perceived exertion that were significantly higher than the remaining firmer surfaces. Previous work has described a rating of ~35 on the CR-100 scale to correspond with the anaerobic threshold in active young adults [24]. Importantly, the *HC* and *HCBB* surfaces, elicited levels of effort closer to the

**Table 3. Heart rate and perceived exertion in response to chest compressions.**

| | | (a)Floor | | | (b)LC | | | (c)LCBB | | | (d)HC | | | (e)HCBB | | |
|---|---|---|---|---|---|---|---|---|---|---|---|---|---|---|---|---|
| **HR** | *Pre (bpm)* | **79** | ± | **13** | **78** | ± | **16** | **80** | ± | **15** | **80** | ± | **15** | **77** | ± | **10** |
| | *CI (95%)* | 74 | - | 85 | 72 | - | 84 | 74 | - | 86 | 74 | - | 86 | 73 | - | 81 |
| | *Post (bpm)* | **125** | ± | **25** [#†] | **128** | ± | **24** | **126** | ± | **25** [#†] | **134** | ± | **25** | **135** | ± | **23** |
| | *CI (95%)* | 114 | - | 136 | 119 | - | 138 | 116 | - | 136 | 124 | - | 144 | 126 | - | 145 |
| | *Sig.* | | < 0.001 | | | < 0.001 | | | < 0.001 | | | < 0.001 | | | < 0.001 | |
| **RPE** | *Pre* | **5.9** | ± | **7.9** | **4.2** | ± | **6.3** | **4.2** | ± | **5.3** | **6.2** | ± | **7.8** | **5.2** | ± | **8.7** |
| | *CI (95%)* | 2.7 | - | 9.1 | 1.6 | - | 6.7 | 2.0 | - | 6.3 | 3.1 | - | 9.3 | 1.7 | - | 8.7 |
| | *Post* | **34.3** | ± | **20.7** [d,e†] | **34.6** | ± | **20.1** [d,e] | **37.4** | ± | **23.3** [d,e#†] | **50.7** | ± | **27.3** [a,b,c] | **46.3** | ± | **23.1** [a,b,c] |
| | *CI (95%)* | 26.0 | - | 42.7 | 26.5 | - | 42.8 | 28.0 | - | 46.8 | 39.5 | - | 62.0 | 37.0 | - | 55.6 |
| | *Sig.* | | < 0.001 | | | < 0.001 | | | < 0.001 | | | < 0.001 | | | < 0.001 | |

Data are mean ± *SD* (n = 27). Floor = concrete floor; LC = low-compliance foam; LCBB = low-compliance foam with backboard; HC = high-compliance foam; HCBB = high compliance foam with backboard. HR = Heart rate. bpm = beats per minute; RPE = CR-100 perceived exertion scale. *Sig.* = significance level pre- versus post-compressions. CI (95%) = 95% confidence interval.

[#]Significant interaction with HC.

[†]Significant interaction with HCBB.

[a]Significantly different versus Floor;

[b]Significantly different versus LC;

[c]Significantly different versus LCBB;

[d]Significantly different versus HC;

[e]Significantly different versus HCBB.

Alpha level = <0.05.

"Heavy" threshold (rating of 50 on the CR-100 scale; [11]). This higher exertion might be attributed to greater mechanical work associated with increased total vertical hand movement, as suggested by Noordergraaf et al. [10], and the elevated muscular effort required to stabilize the lower body on a compliant surface [9]. Our findings suggest that chest compressions upon high-compliance sports matting, may be associated with considerable physiological strain, limiting the duration for which effective CPR can be sustained, even by experienced providers. Such a scenario may, therefore, require more frequent rescuer changeovers during CPR.

Finally, five participants were excluded from the analysis owing to a failure to attain the pre-defined mean compression depth on a solid floor. All were female and of a significantly lower body mass, height, and BMI compared to the remaining female participants (n = 14). Previous work has shown a positive relationship between body mass and stature in CPR efficacy [25]. What remains unclear is whether a more complaint surface may exacerbate this interaction between anthropometry and chest compression quality and warrants future investigation. It may be worth noting that inclusion of these participants in the analysis on an intention-to-treat basis did not alter our conclusions.

Sporting environments are considered higher-risk locations for cardiac arrest [5]. Yet the probability of early recognition of such an event, instigation of high-quality, bystander-delivered CPR, and greater defibrillator access all likely contribute to the superior outcomes observed following cardiac arrest in sporting venues [4, 6, 7]. Our findings highlight additional factors that may be modified or prepared for in order to improve the quality of CPR delivery in such environments. Although further research is required, particularly examining a broader range of matting compliance levels, future guidance should consider the influence of compliant safety matting upon chest compression depth and rescuer fatigue. Where appropriate,

these effects should be acknowledged by medical personnel and also addressed during site-specific CPR training courses for sports venue staff.

## Limitations

Firstly, we tested safety matting comprised of only two compliance levels and thus are unlikely to represent the entire range of sports protection surfaces. Nevertheless, the two matting types assessed were felt to be representative of those commonly encountered in most venues by the research team, participants and gymnasium staff. Moreover, the use of backboards gave our study an additional level of insight as to the effects of surface compliance on chest compression quality.

Secondly, while every effort was made to blind our participants to the surfaces encountered, it is probable that most were able to detect some difference between at least matting, if not backboard conditions. In any event, if our participants were not effectively blinded, it may be that the magnitude of the decline in compression quality was underestimated.

Finally, we recruited a relatively homogenous group of highly-trained and experienced participants as an attempt to control for influences outside of the matting characteristics alone. The addition of data relating to trained bystanders or gymnasium staff may have been beneficial, particularly as these groups are likely to deliver the initial response to a collapsed patient while awaiting trained medical assistance.

## Conclusions

We report a detrimental influence of high-compliance safety matting upon chest compression quality and rescuer exertion during simulated CPR in a sporting environment. This information may be used to guide improved CPR quality upon similar surfaces and may be particularly relevant within venues featuring large areas of compliant matting. These effects should be highlighted to first responders, medical staff and those working within sporting environments. Future research should aim to evaluate the clinical relevance of the decrement imposed by high-compliance matting and the influences of a broader range of surfaces. Moreover, the influence of a CPR backboard and the interactions among surface-type, rescuer training and experience level warrant further investigation.

## Supporting information

**S1 File. Compression decay and degree of chest leaning.** Additional results relating to magnitude of compression decay and chest leaning present upon each surface.
(DOCX)

**S1 Table. Individual participant descriptives.** Participant descriptive data for all included and excluded participants.
(DOCX)

**S1 Data.**
(XLSX)

## Acknowledgments

We wish to thank Richard Davies and Mark Cook at Laerdal Medical for their help and technical support, and Laerdal Medical UK for the loan of equipment. We are also grateful to Climb Newcastle Ltd. for the use of their venue, and participants from the North East Ambulance Service and NHS Hospital Trusts for giving their time to participate.

## Author Contributions

**Conceptualization:** Thomas Kingston, Nicholas B. Tiller, Mukhtar Ahmed, Gareth Jones, Mark I. Johnson, Nigel A. Callender.

**Data curation:** Elle Partington, Gareth Jones, Nigel A. Callender.

**Formal analysis:** Thomas Kingston, Elle Partington, Nigel A. Callender.

**Funding acquisition:** Nigel A. Callender.

**Investigation:** Thomas Kingston, Elle Partington, Gareth Jones, Nigel A. Callender.

**Methodology:** Thomas Kingston, Nicholas B. Tiller, Mukhtar Ahmed, Gareth Jones, Mark I. Johnson, Nigel A. Callender.

**Project administration:** Thomas Kingston, Elle Partington, Nigel A. Callender.

**Resources:** Thomas Kingston, Gareth Jones, Nigel A. Callender.

**Supervision:** Gareth Jones, Mark I. Johnson, Nigel A. Callender.

**Visualization:** Thomas Kingston, Nicholas B. Tiller, Nigel A. Callender.

**Writing – original draft:** Thomas Kingston, Nicholas B. Tiller, Elle Partington, Mukhtar Ahmed, Nigel A. Callender.

**Writing – review & editing:** Thomas Kingston, Nicholas B. Tiller, Elle Partington, Mukhtar Ahmed, Gareth Jones, Mark I. Johnson, Nigel A. Callender.

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
