## [Decision Letter · Decision Letter 0]

25 Mar 2021

PONE-D-21-04446

Sports safety matting diminishes cardiopulmonary resuscitation quality and increases rescuer perceived exertion

PLOS ONE

Dear Dr. Callender,

Thank you for submitting your manuscript to PLOS ONE. After careful consideration, we feel that it has merit but does not fully meet PLOS ONE’s publication criteria as it currently stands. Therefore, we invite you to submit a revised version of the manuscript that addresses the points raised during the review process.

We look forward to receiving your revised manuscript.

Kind regards,

Prof. Ritesh G. Menezes, M.B.B.S., M.D., Diplomate N.B.

Academic Editor

PLOS ONE

Journal Requirements:

'This study received the temporary loan of equipment from Laerdal Medical UK. NAC is

the owner of a commercial indoor climbing gymnasium. NBT is funded by a

postdoctoral fellowship from the Tobacco-Related Disease Research Program

(TRDRP; award no. T31FT1692). MIJ reports grants from Glaxosmithkline and the

Neuromodulation Society of the United Kingdom and Ireland (NSUKI) including

consultancy fees from TENSCare , outside the submitted work. There are no other

conflicts of interest to declare.

Reviewers' comments:

Reviewer's Responses to Questions

**Comments to the Author**

1. Is the manuscript technically sound, and do the data support the conclusions?

Reviewer #1: Partly

Reviewer #2: Yes

Reviewer #3: Partly

2. Has the statistical analysis been performed appropriately and rigorously? 

Reviewer #1: Yes

Reviewer #2: Yes

Reviewer #3: I Don't Know

3. Have the authors made all data underlying the findings in their manuscript fully available?

Reviewer #1: Yes

Reviewer #2: Yes

Reviewer #3: Yes

4. Is the manuscript presented in an intelligible fashion and written in standard English?

Reviewer #1: Yes

Reviewer #2: Yes

Reviewer #3: Yes

5. Review Comments to the Author

Reviewer #1: Mutliple objectives has been reported in the study, however the abstract doesnt mention about CPR preparedness

There is no limitation mentioned in the manuscript

Study has multiple objectives and includes a combination of research and survey. I would recommend reporting survey as separate manuscript and keeping objective simple

The survey portion of the study is definitely worth reporting but may be as research letter rather than original investigation.

Reviewer #2: I would like to thank the authors on their work to put this study and manuscript together. I enjoyed reading the manuscript. The topic and discussion of CPR surfaces is needed and specifically for environments such as sports venues. Below you can find my comments:

1- Overall challenge in the manuscript is it seems there are two main questions/themes the authors are trying to answer , which might confuse the flow and objective of the manuscript.

a. First one is the matting (fold 1 and 2) in the introduction section.

b. Level of CPR preparedness among commercial indoor climbing gymnasiums.

this caused a little bit of a confusion in the sample as it seems both samples are not the same…. In which it is almost having two studies. Have the authors considered having this manuscript focused on only one part, possibly aim 1 and 2? And have #3 separate paper (although may require more sample)?

2- The authors used the term Advanced Life Support (ALS), is this only CPR course (Basic Life Support by American Heart Association)? Or Advanced Cardiac Life Support). ? is ALS by ERC similar to BLS in AHA or ACLS?. If it was ALS=ACLS, what was the rationale behind selecting ACLS and not BLS/CPR ?

a. Not to mention that in the second study (survey on preparedness), the authors used the term BLS training instead of ALS. using one consistent term would be better.

3- Have the authors thought about using indoor venue staff as subjects instead of recruiting medical personnel , such as critical care, EM, etc? what

4- The study design is a crossover design, meaning subjects did multiple rounds of compressions on various surfaces , wouldn’t that cause some sort of fatigue? Even though they got some rest?

Suggestions:

1- Adding a descriptive table of the subjects characteristics.

2- Adding a study figure that shows exactly what subjects went through and how many were in each group etc. Figure would make it easier to visualize study design to the reader. Procedure section.

Reviewer #3: 1. In my opinion Figure 1 is not strictly required and should be removed from the publication.

2. Actual non-significant p values should be specified, rather than being provided as p>0.05

3. Was re-training attempted in the 5 females excluded from study. One would potentially hypothesize that their technique was inadequate and that re-training may have allowed participation in the study.

4. Interesting that Table 2 describes statistically significant differences. However the 95th % CI overlap for those described as different. Additionally, the means of all parameters described are within 1 SD of the remaining parameters defined. The raw data and statistics should be carefully re-checked.

5. Consider adding the 5 females excluded as an "intention to treat" analysis. Did this make any difference to your reported study findings. (I note that this data has been collected as per the attachment in your ofline data file providd

6. PLOS authors have the option to publish the peer review history of their article (what does this mean?). If published, this will include your full peer review and any attached files.

Reviewer #1: No

Reviewer #2: No

Reviewer #3: No

---

## [Author Response · Author response to Decision Letter 0]

12 Apr 2021

Please see attached Word document: Responses to reviewer comments

---

## [Decision Letter · Decision Letter 1]

29 Jun 2021

PONE-D-21-04446R1

Sports safety matting diminishes cardiopulmonary resuscitation quality and increases rescuer perceived exertion

PLOS ONE

Dear Dr. Callender,

Thank you for submitting your manuscript to PLOS ONE. After careful consideration, we feel that it has merit but does not fully meet PLOS ONE’s publication criteria as it currently stands. Therefore, we invite you to submit a revised version of the manuscript that addresses the points raised during the review process.

Please submit your revised manuscript by 06-July-2021. Please include the following items when submitting your revised manuscript:

We look forward to receiving your revised manuscript.

Kind regards,

Prof. Ritesh G. Menezes, M.B.B.S., M.D., Diplomate N.B.

Academic Editor

PLOS ONE

Journal Requirements:

Reviewers' comments:

Reviewer's Responses to Questions

**Comments to the Author**

1. If the authors have adequately addressed your comments raised in a previous round of review and you feel that this manuscript is now acceptable for publication, you may indicate that here to bypass the “Comments to the Author” section, enter your conflict of interest statement in the “Confidential to Editor” section, and submit your "Accept" recommendation.

Reviewer #1: (No Response)

Reviewer #2: All comments have been addressed

Reviewer #3: All comments have been addressed

2. Is the manuscript technically sound, and do the data support the conclusions?

Reviewer #1: Yes

Reviewer #2: Yes

Reviewer #3: Yes

3. Has the statistical analysis been performed appropriately and rigorously? 

Reviewer #1: Yes

Reviewer #2: Yes

Reviewer #3: I Don't Know

4. Have the authors made all data underlying the findings in their manuscript fully available?

Reviewer #1: Yes

Reviewer #2: Yes

Reviewer #3: Yes

5. Is the manuscript presented in an intelligible fashion and written in standard English?

Reviewer #1: Yes

Reviewer #2: Yes

Reviewer #3: Yes

6. Review Comments to the Author

Reviewer #1: My previous concern about multiple objectives in a paper (including survey and research data together) has been resolved by the authors. However the opening line in the discussion section still mentions about "also describe current levels of preparedness for CPR delivery among 222 indoor climbing gymnasiums" Please change this to accurately reflect the current objective of the paper.

Reviewer #2: (No Response)

Reviewer #3: Thank you for addressing my previous comments. I do not have further questions or comments.

7. PLOS authors have the option to publish the peer review history of their article (what does this mean?). If published, this will include your full peer review and any attached files.

Reviewer #1: No

Reviewer #2: No

Reviewer #3: No

---

## [Author Response · Author response to Decision Letter 1]

29 Jun 2021

Thank you again to the Reviewers for their time, comments and suggestions in relation to our paper. Specific responses below.

Reviewer 1: My previous concern about multiple objectives in a paper (including survey and research data together) has been resolved by the authors. However the opening line in the discussion section still mentions about "also describe current levels of preparedness for CPR delivery among 222 indoor climbing gymnasiums" Please change this to accurately reflect the current objective of the paper.

Response: Thank you for highlighting this errant line within the manuscript (Line 221). It has now been removed and the manuscript re-checked for any further outstanding issues. Thank you also for your time reviewing our work. 

Reviewer 2: No comments.

Response: Thank you for your time and previous comments relating to our manuscript.

Reviewer 3: Thank you for addressing my previous comments. I do not have further questions or comments.

Response: Thank you for your time and help in improving our manuscript.

---

## [Editor Report · Decision Letter 2]

5 Jul 2021

Sports safety matting diminishes cardiopulmonary resuscitation quality and increases rescuer perceived exertion

PONE-D-21-04446R2

Dear Dr. Callender,

We’re pleased to inform you that your manuscript has been judged scientifically suitable for publication and will be formally accepted for publication once it meets all outstanding technical requirements.

Kind regards,

Prof. Ritesh G. Menezes, M.B.B.S., M.D., Diplomate N.B.

Academic Editor

PLOS ONE

---

## [Editor Report · Acceptance letter]

13 Jul 2021

PONE-D-21-04446R2 

Sports safety matting diminishes cardiopulmonary resuscitation quality and increases rescuer perceived exertion 

Dear Dr. Callender:

I'm pleased to inform you that your manuscript has been deemed suitable for publication in PLOS ONE. Congratulations! Your manuscript is now with our production department. 

Kind regards, 

on behalf of

Prof. Dr. Ritesh G. Menezes 

Academic Editor

PLOS ONE